# Efficient Compressive Phase Retrieval with Constrained Sensing Vectors

**Sohail Bahmani,   Justin Romberg**
School of Electrical and Computer Engineering.
Georgia Institute of Technology
Atlanta, GA 30332
{sohail.bahmani,jrom}@ece.gatech.edu

## Abstract

We propose a robust and efficient approach to the problem of compressive phase retrieval in which the goal is to reconstruct a sparse vector from the magnitude of a number of its linear measurements. The proposed framework relies on constrained sensing vectors and a two-stage reconstruction method that consists of two standard convex programs that are solved sequentially.

In recent years, various methods are proposed for compressive phase retrieval, but they have suboptimal sample complexity or lack robustness guarantees. The main obstacle has been that there is no straightforward convex relaxations for the type of structure in the target. Given a set of underdetermined measurements, there is a standard framework for recovering a sparse matrix, and a standard framework for recovering a low-rank matrix. However, a general, efficient method for recovering a jointly sparse and low-rank matrix has remained elusive.

Deviating from the models with generic measurements, in this paper we show that if the sensing vectors are chosen at random from an incoherent subspace, then the low-rank and sparse structures of the target signal can be effectively decoupled. We show that a recovery algorithm that consists of a low-rank recovery stage followed by a sparse recovery stage will produce an accurate estimate of the target when the number of measurements is $\mathsf{O}(k \log \frac{d}{k})$, where $k$ and $d$ denote the sparsity level and the dimension of the input signal. We also evaluate the algorithm through numerical simulation.

## 1   Introduction

### 1.1   Problem setting

The problem of *Compressive Phase Retrieval* (CPR) is generally stated as the problem of estimating a $k$-sparse vector $\boldsymbol{x}^\star \in \mathbb{R}^d$ from noisy measurements of the form

$$y_i = |\langle \boldsymbol{a}_i, \boldsymbol{x}^\star \rangle|^2 + z_i \tag{1}$$

for $i = 1, 2, \ldots, n$, where $\boldsymbol{a}_i$ is the sensing vector and $z_i$ denotes the additive noise. In this paper, we study the CPR problem with specific sensing vectors $\boldsymbol{a}_i$ of the form

$$\boldsymbol{a}_i = \boldsymbol{\Psi}^\mathsf{T} \boldsymbol{w}_i, \tag{2}$$

where $\boldsymbol{\Psi} \in \mathbb{R}^{m \times d}$ and $\boldsymbol{w}_i \in \mathbb{R}^m$ are known. In words, the measurement vectors live in a fixed low-dimensional subspace (i.e, the row space of $\boldsymbol{\Psi}$). These types of measurements can be applied in imaging systems that have control over how the scene is illuminated; examples include systems that use structured illumination with a spatial light modulator or a *scattering medium* [1, 2].

By a standard lifting of the signal $\boldsymbol{x}^\star$ to $\boldsymbol{X}^\star = \boldsymbol{x}^\star \boldsymbol{x}^{\star\mathsf{T}}$, the quadratic measurements (1) can be expressed as

$$y_i = \left\langle \boldsymbol{a}_i \boldsymbol{a}_i^\mathsf{T}, \boldsymbol{X}^\star \right\rangle + z_i = \left\langle \boldsymbol{\Psi}^\mathsf{T} \boldsymbol{w}_i \boldsymbol{w}_i^\mathsf{T} \boldsymbol{\Psi}, \boldsymbol{X}^\star \right\rangle + z_i. \tag{3}$$

With the linear operator $\mathcal{W}$ and $\mathcal{A}$ defined as

$$\mathcal{W}: \boldsymbol{B} \mapsto \left[ \left\langle \boldsymbol{w}_i \boldsymbol{w}_i^\mathsf{T}, \boldsymbol{B} \right\rangle \right]_{i=1}^n \qquad \text{and} \qquad \mathcal{A}: \boldsymbol{X} \mapsto \mathcal{W}\left( \boldsymbol{\Psi} \boldsymbol{X} \boldsymbol{\Psi}^\mathsf{T} \right),$$

we can write the measurements compactly as

$$\boldsymbol{y} = \mathcal{A}\left( \boldsymbol{X}^\star \right) + \boldsymbol{z}.$$

Our goal is to estimate the sparse, rank-one, and positive semidefinite matrix $\boldsymbol{X}^\star$ from the measurements (3), which also solves the CPR problem and provides an estimate for the sparse signal $\boldsymbol{x}^\star$ up to the inevitable global phase ambiguity.

**Assumptions** We make the following assumptions throughout the paper.

**A1.** The vectors $\boldsymbol{w}_i$ are independent and have the standard Gaussian distribution on $\mathbb{R}^m$: $\boldsymbol{w}_i \sim \mathsf{N}\left( \boldsymbol{0}, \boldsymbol{I} \right)$.

**A2.** The matrix $\boldsymbol{\Psi}$ is a *restricted isometry* matrix for $2k$-sparse vectors and for a constant $\delta_{2k} \in [0, 1]$. Namely, it obeys

$$(1 - \delta_{2k}) \|\boldsymbol{x}\|_2^2 \leq \|\boldsymbol{\Psi}\boldsymbol{x}\|_2^2 \leq (1 + \delta_{2k}) \|\boldsymbol{x}\|_2^2, \tag{4}$$

for all $2k$-sparse vectors $\boldsymbol{x} \in \mathbb{R}^d$.

**A3.** The noise vector $\boldsymbol{z}$ is bounded as $\|\boldsymbol{z}\|_2 \leq \varepsilon$.

As will be seen in Theorem 1 and its proof below, the Gaussian distribution imposed by the assumption A1 will be used merely to guarantee successful estimation of a rank-one matrix through trace norm minimization. However, other distributions (e.g., uniform distribution on the unit sphere) can also be used to obtain similar guarantees. Furthermore, the restricted isometry condition imposed by the assumption A2 is not critical and can be replaced by weaker assumptions. However, the guarantees obtained under these weaker assumptions usually require more intricate derivations, provide weaker noise robustness, and often do not hold uniformly for all potential target signals. Therefore, to keep the exposition simple and straightforward we assume (4) which is known to hold (with high probability) for various ensembles of random matrices (e.g., Gaussian, Rademacher, partial Fourier, etc). Because in many scenarios we have the flexibility of selecting $\boldsymbol{\Psi}$, the assumption (4) is realistic as well.

**Notation** Let us first set the notation used throughout the paper. Matrices and vectors are denoted by bold capital and small letters, respectively. The set of positive integers less than or equal to $n$ is denoted by $[n]$. The notation $f = \mathsf{O}(g)$ is used when $f = cg$ for some absolute constant $c > 0$. For any matrix $\boldsymbol{M}$, the Frobenius norm, the nuclear norm, the entrywise $\ell_1$-norm, and the largest entrywise absolute value of the entries are denoted by $\|\boldsymbol{M}\|_F$, $\|\boldsymbol{M}\|_*$, $\|\boldsymbol{M}\|_1$, and $\|\boldsymbol{M}\|_\infty$, respectively. To indicate that a matrix $\boldsymbol{M}$ is positive semidefinite we write $\boldsymbol{M} \succeq \boldsymbol{0}$.

## 1.2 Contributions

The main challenge in the CPR problem in its general formulation is to design an accurate estimator that has optimal sample complexity and computationally tractable. In this paper we address this challenge in the special setting where the sensing vectors can be factored as (2). Namely, we propose an algorithm that

- provably produces an accurate estimate of the lifted target $\boldsymbol{X}^\star$ from only $n = \mathsf{O}\left( k \log \frac{d}{k} \right)$ measurements, and

- can be computed in polynomial time through efficient convex optimization methods.

## 1.3 Related work

Several papers including [3, 4, 5, 6, 7] have already studied the application of convex programming for (non-sparse) phase retrieval (PR) in various settings and have established estimation accuracy through different mathematical techniques. These phase retrieval methods attain nearly optimal sample complexities that scales with the dimension of the target signal up to a constant factor [4, 5, 6] or at most a logarithmic factor [3]. However, to the best of our knowledge, the exiting methods for CPR either lack accuracy and robustness guarantees or have suboptimal sample complexities.

The problem of recovering a sparse signal from the magnitude of its subsampled Fourier transforms is cast in [8] as an $\ell_1$-minimization with non-convex constraints. While [8] shows that a sufficient number of measurements would grow quadratically in $k$ (i.e., the sparsity of the signal), the numerical simulations suggest that the non-convex method successfully estimates the sparse signal with only about $k \log \frac{d}{k}$ measurements. Another non-convex approach to CPR is considered in [9] which poses the problem as finding a $k$-sparse vector that minimizes the residual error that takes a quartic form. A local search algorithm called GESPAR [10] is then applied to (approximate) the solution to the formulated sparsity-constrained optimization. This approach is shown to be effective through simulations, but it also lacks global convergence or statistical accuracy guarantees. An alternating minimization method for both PR and CPR is studied in [11]. This method is appealing in large scale problems because of computationally inexpensive iterations. More importantly, [11] proposes a specific initialization using which the alternating minimization method is shown to converge linearly in noise-free PR and CPR. However, the number of measurements required to establish this convergence is effectively quadratic in $k$. In [12] and [13] the $\ell_1$-regularized form of the trace minimization

$$\underset{\boldsymbol{X} \succcurlyeq \boldsymbol{0}}{\operatorname{argmin}} \quad \operatorname{trace}(\boldsymbol{X}) + \lambda \|\boldsymbol{X}\|_1$$
$$\text{subject to} \quad \mathcal{A}(\boldsymbol{X}) = \boldsymbol{y} \tag{5}$$

is proposed for the CPR problem. The guarantees of [13] are based on the restricted isometry property of the sensing operator $\boldsymbol{X} \mapsto [\langle \boldsymbol{a}_i \boldsymbol{a}_i^*, \boldsymbol{X} \rangle]_{i=1}^n$ for sparse matrices. In [12], however, the analysis is based on construction of a *dual certificate* through an adaptation of the *golfing scheme* [14]. Assuming standard Gaussian sensing vectors $\boldsymbol{a}_i$ and with appropriate choice of the regularization parameter $\lambda$, it is shown in [12] that (5) solves the CPR when $n = \mathsf{O}(k^2 \log d)$. Furthermore, this method fails to recover the target sparse and rank-one matrix if $n$ is dominated by $k^2$. Estimation of simultaneously structured matrices through convex relaxations similar to (5) is also studied in [15] where it is shown that these methods do not attain optimal sample complexity. More recently, assuming that the sparse target has a Bernoulli-Gaussian distribution, a *generalized approximate message passing* framework is proposed in [16] to solve the CPR problem. Performance of this method is evaluated through numerical simulations for standard Gaussian sensing matrices which show the *empirical phase transition* for successful estimation occurs at $n = \mathsf{O}(k \log \frac{d}{k})$ and also the algorithms can have a significantly lower runtime compared to some of the competing algorithms including GESPAR [10] and CPRL [13]. The PhaseCode algorithm is proposed in [17] to solve the CPR problem with sensing vectors designed using sparse graphs and techniques adapted from coding theory. Although PhaseCode is shown to achieve the optimal sample complexity, it lacks robustness guarantees.

While preparing the final version of the current paper, we became aware of [18] which has independently proposed an approach similar to ours to address the CPR problem.

## 2 Main Results

### 2.1 Algorithm

We propose a two-stage algorithm outlined in Algorithm 1. Each stage of the algorithm is a convex program for which various efficient numerical solvers exists. In the first stage we solve (6) to obtain a low-rank matrix $\widehat{\boldsymbol{B}}$ which is an estimator of the matrix

$$\boldsymbol{B}^\star = \boldsymbol{\Psi} \boldsymbol{X}^\star \boldsymbol{\Psi}^\mathsf{T}.$$

Then $\widehat{B}$ is used in the second stage of the algorithm as the measurements for a sparse estimation expressed by (7). The constraint of (7) depends on an absolute constant $C > 0$ that should be sufficiently large.

---

**Algorithm 1:**

---

    **input**  : the measurements $y$, the operator $\mathcal{W}$, and the matrix $\boldsymbol{\Psi}$

    **output**: the estimate $\widehat{X}$

**1** Low-rank estimation stage:

$$\widehat{B} \in \underset{B \succcurlyeq 0}{\operatorname{argmin}} \quad \operatorname{trace}(B)$$
$$\text{subject to} \quad \|\mathcal{W}(B) - y\|_2 \leq \varepsilon \tag{6}$$

**2** Sparse estimation stage:

$$\widehat{X} \in \underset{X}{\operatorname{argmin}} \quad \|X\|_1$$
$$\text{subject to} \quad \left\|\boldsymbol{\Psi} X \boldsymbol{\Psi}^{\mathsf{T}} - \widehat{B}\right\|_F \leq \frac{C\varepsilon}{\sqrt{n}} \tag{7}$$

---

**Post-processing.** The result of the low-rank estimation stage (6) is generally not rank-one. Similarly, the sparse estimation stage does not necessarily produce a $\widehat{X}$ that is $k \times k$-sparse (i.e., it has at most $k$ nonzero rows and columns) and rank-one. In fact, since we have not imposed the positive semidefiniteness constraint (i.e., $X \succcurlyeq 0$) in (7), the estimate $\widehat{X}$ is not even guaranteed to be positive semidefinite (PSD). However, we can enforce the rank-one or the sparsity structure in postprocessing steps simply by projecting the produced estimate on the set of rank-one or $k \times k$-sparse PSD matrices. The simple but important observation is that projecting $\widehat{X}$ onto the desired sets at most doubles the estimation error. This fact is shown by Lemma 2 in Section 4 in a general setting.

**Alternatives.** There are alternative convex relaxations for the low-rank estimation and the sparse estimation stages of Algorithm (1). For example, (6) can be replaced by its regularized least squares analog

$$\widehat{B} \in \underset{B \succcurlyeq 0}{\operatorname{argmin}} \quad \frac{1}{2}\|\mathcal{W}(B) - y\|_2^2 + \lambda \|B\|_* \,,$$

for an appropriate choice of the regularization parameter $\lambda$. Similarly, instead of (7) we can use an $\ell_1$-regularized least squares. Furthermore, to perform the low-rank estimation and the sparse estimation we can use non-convex greedy type algorithms that typically have lower computational costs. For example, the low-rank estimation stage can be performed via the Wirtinger flow method proposed in [19]. Furthermore, various greedy compressive sensing algorithms such as the Iterative Hard Thresholding [20] and CoSaMP [21] can be used to solve the desired sparse estimation. To guarantee the accuracy of these compressive sensing algorithms, however, we might need to adjust the assumption A2 to have the restricted isometry property for $ck$-sparse vectors with $c$ being some small positive integer.

## 2.2 Accuracy guarantees

The following theorem shows that any solution of the proposed algorithm is an accurate estimator of $X^\star$.

**Theorem 1.** *Suppose that the assumptions A1, A2, and A3 hold with a sufficiently small constant* $\delta_{2k}$. *Then, there exist positive absolute constants* $C_1$, $C_2$, *and* $C_3$ *such that if*

$$n \geq C_1 m, \tag{8}$$

*then any estimate* $\widehat{X}$ *of the Algorithm 1 obeys*

$$\left\|\widehat{X} - X^\star\right\|_F \leq \frac{C_2 \varepsilon}{\sqrt{n}},$$

*for all rank-one and $k \times k$-sparse matrices $\boldsymbol{X}^\star \succcurlyeq \boldsymbol{0}$ with probability exceeding $1 - e^{-C_3 n}$.*

The proof of Theorem 1 is straightforward and is provided in Section 4. The main idea is first to show the low-rank estimation stage produces an accurate estimate of $\boldsymbol{B}^\star$. Because this stage can be viewed as a standard phase retrieval through lifting, we can simply use accuracy guarantees that are already established in the literature (e.g., [3, 6, 5]). In particular, we use [5, Theorem 2] which established an error bound that holds uniformly for all valid $\boldsymbol{B}^\star$. Thus we can ensure that $\boldsymbol{X}^\star$ is feasible in the sparse estimation stage. Then the accuracy of the sparse estimation stage can also be established by a simple adaptation of the analyses based on the restricted isometry property such as [22].

The dependence of $n$ (i.e., the number of measurements) and $k$ (i.e., the sparsity of the signal) is not explicit in Theorem 1. This dependence is absorbed in $m$ which must be sufficiently large for Assumption A2 to hold. Considering a Gaussian matrix $\boldsymbol{\Psi}$, the following corollary gives a concrete example where the dependence of $n$ on $k$ through $m$ is exposed.

**Corollary 1.** *Suppose that the assumptions of Theorem 1 including (8) hold. Furthermore, suppose that $\boldsymbol{\Psi}$ is a Gaussian matrix with iid $\mathsf{N}\left(0, \frac{1}{m}\right)$ entries and*

$$m \geq c_1 k \, \log \frac{d}{k}, \tag{9}$$

*for some absolute constant $c_1 > 0$. Then any estimate $\widehat{\boldsymbol{X}}$ produced by Algorithm 1 obeys*

$$\left\| \widehat{\boldsymbol{X}} - \boldsymbol{X}^\star \right\|_F \leq \frac{C_2 \varepsilon}{\sqrt{n}},$$

*for all rank-one and $k \times k$-sparse matrices $\boldsymbol{X}^\star \succcurlyeq \boldsymbol{0}$ with probability exceeding $1 - 3e^{-c_2 m}$ for some constant $c_2 > 0$.*

*Proof.* It is well-known that if $\boldsymbol{\Psi}$ has iid $\mathsf{N}\left(0, \frac{1}{m}\right)$ and we have (9) then (4) holds with high probability. For example, using a standard covering argument and a union bound [23] shows that if (9) holds for a sufficiently large constant $c_1 > 0$ then we have (4) for a sufficiently small constant $\delta_{2k}$ with probability exceeding $1 - 2e^{-cm}$ for some constant $c > 0$ that depends only on $\delta_{2k}$. Therefore, Theorem 1 yields the desired result which holds with probability exceeding $1 - 2e^{-cm} - e^{-C_3 n} \geq 1 - 3e^{-c_2 m}$ for some constant $c_2 > 0$ depending only on $\delta_{2k}$. □

## 3 Numerical Experiments

We evaluated the performance of Algorithm 1 through some numerical simulations. The low-rank estimation stage and the sparse estimation stage are implemented using the TFOCS package [24]. We considered the target $k$-sparse signal $\boldsymbol{x}^\star$ to be in $\mathbb{R}^{256}$ (i.e., $d = 256$). The support set of of the target signal is selected uniformly at random and the entry values on this support are drawn independently from $\mathsf{N}(0, 1)$. The noise vector $\boldsymbol{z}$ is also Gaussian with independent $\mathsf{N}\left(0, 10^{-4}\right)$. The operator $\mathcal{W}$ and the matrix $\boldsymbol{\Psi}$ are drawn from some Gaussian ensembles as described in Corollary 1. We measured the relative error $\frac{\|\widehat{\boldsymbol{X}} - \boldsymbol{X}^\star\|_F}{\|\boldsymbol{X}^\star\|_F}$ of achieved by the compared methods over 100 trials with sparsity level (i.e., $k$) varying in the set $\{2, 4, 6, \dots, 20\}$.

In the first experiment, for each value of $k$, the pair $(m, n)$ that determines the size $\mathcal{W}$ and $\boldsymbol{\Psi}$ are selected from $\{(8k, 24k), (8k, 32k), (12k, 36k), (12k, 48k), (16k, 48k)\}$. Figure 1 illustrates the 0.9 quantiles of the relative error versus $k$ for the mentioned choices of $m$.

In the second experiment we compared the performance of Algorithm 1 to the convex optimization methods that do not exploit the structure of the sensing vectors. The setup for this experiment is the same as in the first experiment except for the size of $\mathcal{W}$ and $\boldsymbol{\Psi}$; we chose $m = \left\lceil 2k \left(1 + \log \frac{d}{k}\right) \right\rceil$ and $n = 3m$, where $\lceil r \rceil$ denotes the smallest integer greater than $r$. Figure 2 illustrates the 0.9 quantiles of the measured relative errors for Algorithm 1, the semidefinite program (5) for $\lambda = 0$ and $\lambda = 0.2$, and the $\ell_1$-minimization

$$\begin{aligned} \underset{\boldsymbol{X}}{\text{argmin}} \quad & \|\boldsymbol{X}\|_1 \\ \text{subject to} \quad & \mathcal{A}(\boldsymbol{X}) = \boldsymbol{y}, \end{aligned}$$

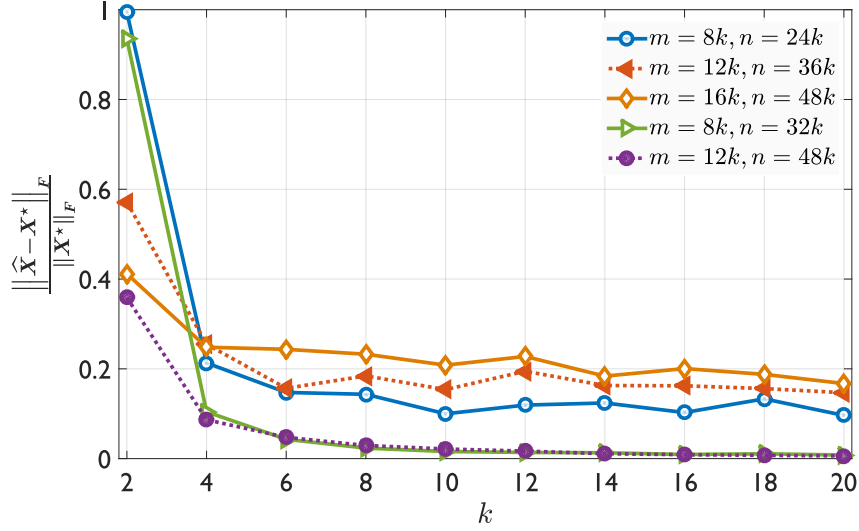

Figure 1: The empirical 0.9 quantile of the relative estimation error vs. sparsity for various choices of $m$ and $n$ with $d = 256$.

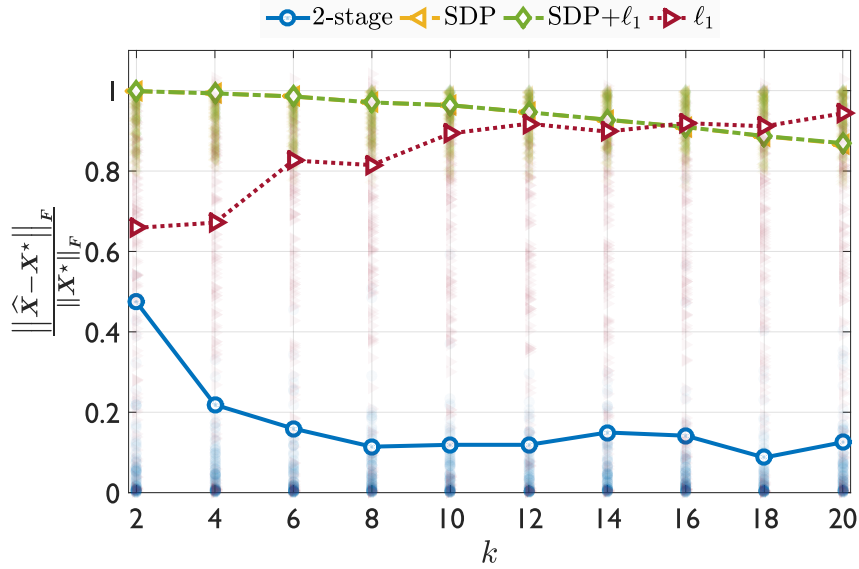

Figure 2: The empirical 0.9 quantile of the relative estimation error vs. sparsity for Algorithm 1 and different trace- and/or $\ell_1$- minimization methods with $d = 256$, $m = \left\lceil 2k \left(1 + \log \frac{d}{k}\right)\right\rceil$, and $n = 3m$.

which are denoted by 2-stage, SDP, SDP+$\ell_1$, and $\ell_1$, respectively. The SDP-based method did not perform significantly different for other values of $\lambda$ in our complementary simulations. The relative error for each trial is also overlaid in Figure 2 visualize its empirical distribution. The empirical performance of the algorithms are in agreement with the theoretical results. Namely in a regime where $n = O(m) = O\left(k \log \frac{d}{k}\right)$, Algorithm 1 can produce accurate estimates whereas while the other approaches fail in this regime. The SDP and SDP+$\ell_1$ show nearly identical performance. The $\ell_1$-minimization, however, competes with Algorithm 1 for small values of $k$. This observation can be explained intuitively by the fact that the $\ell_1$-minimization succeeds with $n = O\left(k^2\right)$ measurements which for small values of $k$ can be sufficiently close to the considered $n = 3\left\lceil 2k \left(1 + \log \frac{d}{k}\right)\right\rceil$ measurements.

# 4 Proofs

*Proof of Theorem 1.* Clearly, $\boldsymbol{B}^\star = \boldsymbol{\Psi}\boldsymbol{X}^\star\boldsymbol{\Psi}^\mathsf{T}$ is feasible in 6 because of A3. Therefore, we can show that any solution $\widehat{\boldsymbol{B}}$ of (6) accurately estimates $\boldsymbol{B}^\star$ using existing results on nuclear-norm minimization. In particular, we can invoke [5, Theorem 2 and Section 4.3] which guarantees that for some positive absolute constants $C_1$, $C_2'$, and $C_3$ if (8) holds then

$$\left\|\widehat{\boldsymbol{B}} - \boldsymbol{B}^\star\right\|_F \le \frac{C_2'\varepsilon}{\sqrt{n}},$$

holds for all valid $\boldsymbol{B}^\star$, thereby for all valid $\boldsymbol{X}^\star$, with probability exceeding $1 - e^{-C_3 n}$. Therefore, with $C = C_2'$, the target matrix $\boldsymbol{X}^\star$ would be feasible in (7). Now, it suffices to show that the sparse estimation stage can produce an accurate estimate of $\boldsymbol{X}^\star$. Recall that by A2, the matrix $\boldsymbol{\Psi}$ is restricted isometry for $2k$-sparse vectors. Let $\boldsymbol{X}$ be a matrix that is $2k \times 2k$-sparse, i.e., a matrix whose entries except for some $2k \times 2k$ submatrix are all zeros. Applying (4) to the columns of $\boldsymbol{X}$ and adding the inequalities yield

$$(1 - \delta_{2k})\|\boldsymbol{X}\|_F^2 \le \|\boldsymbol{\Psi}\boldsymbol{X}\|_F^2 \le (1 + \delta_{2k})\|\boldsymbol{X}\|_F^2. \tag{10}$$

Because the columns of $\boldsymbol{X}^\mathsf{T}\boldsymbol{\Psi}^\mathsf{T}$ are also $2k$-sparse we can repeat the same argument and obtain

$$(1 - \delta_{2k})\left\|\boldsymbol{X}^\mathsf{T}\boldsymbol{\Psi}^\mathsf{T}\right\|_F^2 \le \left\|\boldsymbol{\Psi}\boldsymbol{X}^\mathsf{T}\boldsymbol{\Psi}^\mathsf{T}\right\|_F^2 \le (1 + \delta_{2k})\left\|\boldsymbol{X}^\mathsf{T}\boldsymbol{\Psi}^\mathsf{T}\right\|_F^2. \tag{11}$$

Using the facts that $\left\|\boldsymbol{X}^\mathsf{T}\boldsymbol{\Psi}^\mathsf{T}\right\|_F = \|\boldsymbol{\Psi}\boldsymbol{X}\|_F$ and $\left\|\boldsymbol{\Psi}\boldsymbol{X}^\mathsf{T}\boldsymbol{\Psi}^\mathsf{T}\right\|_F = \left\|\boldsymbol{\Psi}\boldsymbol{X}\boldsymbol{\Psi}^\mathsf{T}\right\|_F$, the inequalities (10) and (11) imply that

$$(1 - \delta_{2k})^2\|\boldsymbol{X}\|_F^2 \le \left\|\boldsymbol{\Psi}\boldsymbol{X}\boldsymbol{\Psi}^\mathsf{T}\right\|_F^2 \le (1 + \delta_{2k})^2\|\boldsymbol{X}\|_F^2. \tag{12}$$

The proof proceeds with an adaptation of the arguments used to prove accuracy of $\ell_1$-minimization in compressive sensing based on the restricted isometry property (see, e.g., [22]). Let $\boldsymbol{E} = \widehat{\boldsymbol{X}} - \boldsymbol{X}^\star$. Furthermore, let $S_0 \subseteq [d] \times [d]$ denote the support set of the $k \times k$-sparse target $\boldsymbol{X}^\star$. Define $\boldsymbol{E}_0$ to be a $d \times d$ matrix that is identical to $\boldsymbol{E}$ over the index set $S_0$ and zero elsewhere. By optimality of $\widehat{\boldsymbol{X}}$ and feasibility of $\boldsymbol{X}^\star$ in (7) we have

$$\|\boldsymbol{X}^\star\|_1 \ge \left\|\widehat{\boldsymbol{X}}\right\|_1 = \|\boldsymbol{X}^\star + \boldsymbol{E} - \boldsymbol{E}_0 + \boldsymbol{E}_0\|_1 \ge \|\boldsymbol{X}^\star + \boldsymbol{E} - \boldsymbol{E}_0\|_1 - \|\boldsymbol{E}_0\|_1$$
$$= \|\boldsymbol{X}^\star\|_1 + \|\boldsymbol{E} - \boldsymbol{E}_0\|_1 - \|\boldsymbol{E}_0\|_1,$$

where the last line follows from the fact that $\boldsymbol{X}^\star$ and $\boldsymbol{E} - \boldsymbol{E}_0$ have disjoint supports. Thus, we have

$$\|\boldsymbol{E} - \boldsymbol{E}_0\|_1 \le \|\boldsymbol{E}_0\|_1 \le k\|\boldsymbol{E}_0\|_F. \tag{13}$$

Now consider a decomposition of $\boldsymbol{E} - \boldsymbol{E}_0$ as the sum

$$\boldsymbol{E} - \boldsymbol{E}_0 = \sum_{j=1}^J \boldsymbol{E}_j, \tag{14}$$

such that for $j \ge 0$ the $d \times d$ matrices $\boldsymbol{E}_j$ have disjoint support sets of size $k \times k$ except perhaps for the last few matrices that might have smaller supports. More importantly, the partitioning matrices $\boldsymbol{E}_j$ are chosen to have a decreasing Frobenius norm (i.e., $\|\boldsymbol{E}_j\|_F \ge \|\boldsymbol{E}_{j+1}\|_F$) for $j \ge 1$. We have

$$\left\|\sum_{j=2}^J \boldsymbol{E}_j\right\|_F \le \sum_{j=2}^J \|\boldsymbol{E}_j\|_F \le \frac{1}{k}\sum_{j=2}^J \|\boldsymbol{E}_{j-1}\|_1 \le \frac{1}{k}\|\boldsymbol{E} - \boldsymbol{E}_0\|_1 \le \|\boldsymbol{E}_0\|_F \le \|\boldsymbol{E}_0 + \boldsymbol{E}_1\|_F, \tag{15}$$

where the chain of inequalities follow from the triangle inequality, the fact that $\|\boldsymbol{E}_j\|_\infty \le \frac{1}{k^2}\|\boldsymbol{E}_{j-1}\|_1$ by construction, the fact that the matrices $\boldsymbol{E}_j$ have disjoint support and satisfy (14), the bound (13), and the fact that $\boldsymbol{E}_0$ and $\boldsymbol{E}_1$ are orthogonal. Furthermore, we have

$$\left\|\boldsymbol{\Psi}(\boldsymbol{E}_0 + \boldsymbol{E}_1)\boldsymbol{\Psi}^\mathsf{T}\right\|_F^2 = \left\langle \boldsymbol{\Psi}(\boldsymbol{E}_0 + \boldsymbol{E}_1)\boldsymbol{\Psi}^\mathsf{T}, \boldsymbol{\Psi}\left(\boldsymbol{E} - \sum_{j=2}^J \boldsymbol{E}_j\right)\boldsymbol{\Psi}^\mathsf{T}\right\rangle$$

$$\le \left\|\boldsymbol{\Psi}(\boldsymbol{E}_0 + \boldsymbol{E}_1)\boldsymbol{\Psi}^\mathsf{T}\right\|_F \left\|\boldsymbol{\Psi}\boldsymbol{E}\boldsymbol{\Psi}^\mathsf{T}\right\|_F + \sum_{i=0}^1 \sum_{j=2}^J \left|\left\langle \boldsymbol{\Psi}\boldsymbol{E}_i\boldsymbol{\Psi}^\mathsf{T}, \boldsymbol{\Psi}\boldsymbol{E}_j\boldsymbol{\Psi}^\mathsf{T}\right\rangle\right|, \tag{16}$$

where the first term is obtained by the Cauchy-Schwarz inequality and the summation is obtained by the triangle inequality. Because $\boldsymbol{E} = \widehat{\boldsymbol{X}} - \boldsymbol{X}^{\star}$ by definition, the triangle inequality and the fact that $\boldsymbol{X}^{\star}$ and $\widehat{\boldsymbol{X}}$ are feasible in (7) imply that $\left\| \boldsymbol{\Psi} \boldsymbol{E} \boldsymbol{\Psi}^{\mathsf{T}} \right\|_{F} \leq \left\| \boldsymbol{\Psi} \widehat{\boldsymbol{X}} \boldsymbol{\Psi}^{\mathsf{T}} - \widehat{\boldsymbol{B}} \right\|_{F} + \left\| \boldsymbol{\Psi} \boldsymbol{X}^{\star} \boldsymbol{\Psi}^{\mathsf{T}} - \widehat{\boldsymbol{B}} \right\|_{F} \leq \frac{2C\varepsilon}{\sqrt{n}}$. Furthermore, Lemma 1 below which is adapted from [22, Lemma 2.1] guarantees that for $i \in \{0,1\}$ and $j \geq 2$ we have $\left| \left\langle \boldsymbol{\Psi} \boldsymbol{E}_i \boldsymbol{\Psi}^{\mathsf{T}}, \boldsymbol{\Psi} \boldsymbol{E}_j \boldsymbol{\Psi}^{\mathsf{T}} \right\rangle \right| \leq 2\delta_{2k} \left\| \boldsymbol{E}_i \right\|_F \left\| \boldsymbol{E}_j \right\|_F$. Therefore, we obtain

$$
\begin{aligned}
(1 - \delta_{2k})^2 \left\| \boldsymbol{E}_0 + \boldsymbol{E}_1 \right\|_F^2 & \leq \left\| \boldsymbol{\Psi} \left( \boldsymbol{E}_0 + \boldsymbol{E}_1 \right) \boldsymbol{\Psi}^{\mathsf{T}} \right\|_F^2 \\
& \leq \frac{2C\varepsilon}{\sqrt{n}} \left\| \boldsymbol{\Psi} \left( \boldsymbol{E}_0 + \boldsymbol{E}_1 \right) \boldsymbol{\Psi}^{\mathsf{T}} \right\|_F + 2\delta_{2k} \sum_{i=0}^{1} \sum_{j=2}^{J} \left\| \boldsymbol{E}_i \right\|_F \left\| \boldsymbol{E}_j \right\|_F \\
& \leq \frac{2C\varepsilon}{\sqrt{n}} \left( 1 + \delta_{2k} \right) \left\| \boldsymbol{E}_0 + \boldsymbol{E}_1 \right\|_F + 2\delta_{2k} \sum_{i=0}^{1} \sum_{j=2}^{J} \left\| \boldsymbol{E}_i \right\|_F \left\| \boldsymbol{E}_j \right\|_F \\
& \leq \frac{2C\varepsilon}{\sqrt{n}} \left( 1 + \delta_{2k} \right) \left\| \boldsymbol{E}_0 + \boldsymbol{E}_1 \right\|_F + 2\delta_{2k} \left( \left\| \boldsymbol{E}_0 \right\|_F + \left\| \boldsymbol{E}_1 \right\|_F \right) \left\| \boldsymbol{E}_0 + \boldsymbol{E}_1 \right\|_F \\
& \leq \left\| \boldsymbol{E}_0 + \boldsymbol{E}_1 \right\|_F \left( \frac{2C\varepsilon}{\sqrt{n}} \left( 1 + \delta_{2k} \right) + 2\sqrt{2} \delta_{2k} \left\| \boldsymbol{E}_0 + \boldsymbol{E}_1 \right\|_F \right)
\end{aligned}
$$

where the chain of inequalities follow from the lower bound in (12), the bound (16), the upper bound in (12), the bound (15), and the fact that $\left\| \boldsymbol{E}_0 \right\|_F + \left\| \boldsymbol{E}_1 \right\|_F \leq \sqrt{2} \left\| \boldsymbol{E}_0 + \boldsymbol{E}_1 \right\|_F$. If $\delta_{2k} < 1 + \sqrt{2} \left( 1 - \sqrt{1 + \sqrt{2}} \right) \approx 0.216$, then we have $\gamma := (1 - \delta_{2k})^2 - 2\sqrt{2} \delta_{2k} > 0$ and thus

$$
\left\| \boldsymbol{E}_0 + \boldsymbol{E}_1 \right\|_F \leq \frac{2C \left( 1 + \delta_{2k} \right) \varepsilon}{\gamma \sqrt{n}}.
$$

Adding the above inequality to (13) and applying the triangle then yields the desired result. $\square$

**Lemma 1.** *Let $\boldsymbol{\Psi}$ be a matrix obeying (4). Then for any pair of $k \times k$-sparse matrices $\boldsymbol{X}$ and $\boldsymbol{X}'$ with disjoint supports we have*

$$
\left| \left\langle \boldsymbol{\Psi} \boldsymbol{X} \boldsymbol{\Psi}^{\mathsf{T}}, \boldsymbol{\Psi} \boldsymbol{X}' \boldsymbol{\Psi}^{\mathsf{T}} \right\rangle \right| \leq 2\delta_{2k} \left\| \boldsymbol{X} \right\|_F \left\| \boldsymbol{X}' \right\|_F.
$$

*Proof.* Suppose that $\boldsymbol{X}$ and $\boldsymbol{X}'$ have unit Frobenius norm. Using the identity $\left\langle \boldsymbol{\Psi} \boldsymbol{X} \boldsymbol{\Psi}^{\mathsf{T}}, \boldsymbol{\Psi} \boldsymbol{X}' \boldsymbol{\Psi}^{\mathsf{T}} \right\rangle = \frac{1}{4} \left( \left\| \boldsymbol{\Psi} \left( \boldsymbol{X} + \boldsymbol{X}' \right) \boldsymbol{\Psi}^{\mathsf{T}} \right\|_F^2 - \left\| \boldsymbol{\Psi} \left( \boldsymbol{X} - \boldsymbol{X}' \right) \boldsymbol{\Psi}^{\mathsf{T}} \right\|_F^2 \right)$ and the fact that $\boldsymbol{X}$ and $\boldsymbol{X}'$ have disjoint supports, it follows from (12) that

$$
-2\delta_{2k} = \frac{(1 - \delta_{2k})^2 - (1 + \delta_{2k})^2}{2} \leq \left\langle \boldsymbol{\Psi} \boldsymbol{X} \boldsymbol{\Psi}^{\mathsf{T}}, \boldsymbol{\Psi} \boldsymbol{X}' \boldsymbol{\Psi}^{\mathsf{T}} \right\rangle \leq \frac{(1 + \delta_{2k})^2 - (1 - \delta_{2k})^2}{2} = 2\delta_{2k}.
$$

The general result follows immediately as the desired inequality is homogeneous in the Frobenius norms of $\boldsymbol{X}$ and $\boldsymbol{X}'$. $\square$

**Lemma 2** (Projected estimator)**.** *Let $S$ be a closed nonempty subset of a normed vector space $(\mathbb{V}, \|\cdot\|)$. Suppose that for $\boldsymbol{v}^{\star} \in S$ we have an estimator $\widehat{\boldsymbol{v}} \in \mathbb{V}$, not necessarily in $S$, that obeys $\|\widehat{\boldsymbol{v}} - \boldsymbol{v}^{\star}\| \leq \epsilon$. If $\widetilde{\boldsymbol{v}}$ denotes a projection of $\widehat{\boldsymbol{v}}$ onto $S$, then we have $\|\widetilde{\boldsymbol{v}} - \boldsymbol{v}^{\star}\| \leq 2\epsilon$.*

*Proof.* By definition $\widetilde{\boldsymbol{v}} \in \arg\min_{\boldsymbol{v} \in S} \|\boldsymbol{v} - \widehat{\boldsymbol{v}}\|$. Therefore, because $\boldsymbol{v}^{\star} \in S$ we have

$$
\|\widetilde{\boldsymbol{v}} - \boldsymbol{v}^{\star}\| \leq \|\widehat{\boldsymbol{v}} - \boldsymbol{v}^{\star}\| + \|\widetilde{\boldsymbol{v}} - \widehat{\boldsymbol{v}}\| \leq 2 \|\widehat{\boldsymbol{v}} - \boldsymbol{v}^{\star}\| \leq 2\epsilon.
$$

$\square$

## Acknowledgements

This work was supported by ONR grant N00014-11-1-0459, and NSF grants CCF-1415498 and CCF-1422540.

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
