[Reviews · NeurIPS 2015]

Submitted by Assigned_Reviewer_1

The paper presents a method for compressive phase retrieval of sparse vectors. The main idea is to use two phase method for compressive phase retrieval. In the first phase, one uses standard compressive sensing to encode the sparse vector into a low-dimensional vector and then use standard phase sensing style encoding. Hence for recovery, the method first recovers the low-dimensional encoding and then use standard compressive sensing to recover the original sparse signal.

Comments: a) The paper is well written and the result is presented very nicely.

b) The result is stronger than the existing results for compressive phase retrieval.

c) Unfortunately, the main issue with the paper is that the proposed measurement scheme is not very practical and does not have strong practical applications. Typically, phase retrieval is an important problem when the measurements are generated using Fourier transform matrices. Even the standard Gaussian measurement schemes (studied in several recent works) do not have many practical applications. But it is believed to be a good proxy for Fourier measurements, so there is good interest. However, for the proposed scheme, the practical applications are not well known. Moreover, the proposed scheme is not particularly novel and similar two-stage algorithms have been discussed earlier as well (although in context of other compressive sensing problems). Ref: One bit compressive sensing:Provable support and vector recovery by Gopi et al'ICML2013.

d) Authors use the method by Kueng et al'2014 for recovering the compressed vectors in phase one. However, this is not required and one can use the standard phase retrieval methods for the same (for example, phaselifting method). The advantage of that method is that it works even for complex vectors and measurements.

Overall, a nicely written result. But the practical implications of the measurement scheme are fairly limited and the novelty in technique is also fairly limited.

Summary: See below.

Submitted by Assigned_Reviewer_2

Paper proposes a two stage method for noisy compressive phase retrieval. The sensing vectors belong to the range space of some matrix, \Psi, which in this paper is assumed to have IID Gaussian components. The proposed method breaks the problem into two stages In the 1st stage the problem is embedded into a lifted space and a low rank matrix is estimated. Then a sparse approximation to the estimated low rank matrix is obtained. Each of these problems have well-known convex-relaxations with guaranteed approximation bounds. What the authors are able to show is that the two stages when put together leads to guaranteed bounds on the entire problem.

The paper is still a bit mysterious to me even though I have checked the proofs and am quite familiar with the arguments. Nevertheless, I was hoping to understand why the two steps are critical. The paper does not do a good job in articulating the need for the two step algorithm apart from the fact that it happens to work! The authors consider sensing vectors of the form a = \psi w where both \psi and w are drawn from a Gaussian ensemble. The other w is used during the low-rank estimation stage while the \psi matrix is utilized for ensuring isometry required in the sparsification stage. Is it possible to contemplate (\psi)w

as a single entity without the need for this factorization and then create a virtual \psi for the argument to go through? Another question is whether or not a mixed objective that combines L1 and nuclear norms into a single stage could lead to a a similar result.

Separately, it is strange to see that signal-to-noise ratio does not explicitly appear in the bound. For instance if I were to scale w then the error bound should scale with w.

In summary the paper is well written and presents interesting results.
Summary: Paper proposes a method for noisy compressive phase retrieval based on a two-stage process. Each stage is a convex optimization problem that can be solved using standard convex programming methods. While the solution strategy appears to be quite straightforward and is heavily based on existing L1 techniques, what is surprising is that this relatively simple strategy leads to guaranteed bounds. The paper could do a better job of explaining why the two steps are necessary and why it would be difficult to do it in a single step.

Submitted by Assigned_Reviewer_3

In this paper the authors propose a two stage approach to the compressive phase retrieval problem. The idea is that if the measurement matrix factorizes into the product of a Gaussian matrix and and RIP matrix one can use a convex program to recover the RIP matrix times the sparse signal and then use standard techniques to recover the sparse signal from such RIP measurements. The paper text is readable and the results are correct. However, my main concern is that this approach heavily relies on modeling assumptions of the measurement ensemble that do not correspond to a realistic model of interest.

I only have one main comment. (The proofs seem correct to me):

First sentence of the abstract: "We propose a new approach to the problem of compressive phase...", last sentence of the second paragraph, and first sentence of 1.2 "the main challenge in the CPR problem in its general formulation is to design an accurate estimator that has optimal sample complexity and computationally tractable"

I disagree with this claim. If the paper is to be accepted I would recommend some caution about such statements and claims. Indeed the main challenge is to get to optimal sample and computational complexity. However, in this process you are not allowed to pick your algorithm based on your sensing mechanism! This is certainly not elusive. In particular if one can choose the measurement models there are already papers that do get to optimal sample complexity and this one would not be the first. e.g. see

"Fast and Robust Compressive Phase Retrieval with Sparse-Graph Codes"

Dong Yin, Kangwook Lee, Ramtin Pedarsani, and Kannan Ramchandran

Let me clarify what I mean. Imagine you get just generic Gaussian measurement vectors this approach simply can not work because the matrix does not factorize. For example, In a realistic setup you have a signal which is sparse in an over-complete basis and you wish to recover that signal from quadratic measurements of the order of sparsity (say from a few Gaussian or Fourier measurements). That is the overall measurement matrix w.r.t the sparse signal is a fat matrix times a fat matrix. Not a tall matrix multiplied by a fat matrix. So even if the RIP assumption holds on the over-complete dictionary it is still not a useful model. Simply put model (2) is unrealistic.

The challenge is sparse phase retrieval is really can we come up with an algorithm that can handle sparsity o(k) e.g. an algorithm that works generically like for Gaussian measurements.

Summary: The paper text is readable and the results are correct. However, my main concern is that this approach heavily relies on modeling assumptions of the measurement ensemble that does not correspond to a realistic model of interest in applications.

Submitted by Assigned_Reviewer_4

The paper exposes in a very clear manner a simple, but not banal, method to perform compressive phase retrieval. The method is based on two steps, a low-rank estimation stage and a sparse estimation stage. Combining previously provided statistical guarantees for the two stages, the authors can provide one for their approach.

The two stages can seemingly be solved efficiently via convex optimization, however the authors do not empirically corroborate their efficiency statement, nor provide detailed computational complexity considerations.

In my opinion, the numerical results are inconclusive, since the comparison with the semi-definite program (5) is evaluated only for two values of the regularization parameter.

Overall, an interesting approach for CPR which needs to be further matured.
Summary: The theoretical analysis is mostly based on previous work and the numerical results are, in my opinion, inconclusive, since the comparison with the semi-definite program (5) is evaluated only for two values of the regularization parameter. Furthermore, the authors do not empirically corroborate their efficiency statement, nor provide detailed computational complexity considerations.

Author Feedback
Author rebuttal: We would like to thank all reviewers for their constructive comments. We address the common concerns of the reviewers in more details below. Some of the other concerns that require shorter response are addressed at the end.

Relevance of the measurement model:

In the proposed measurement model, the sensing vectors are drawn from a fixed subspace. In our analysis, this subspace corresponds to the row space of a "compressed sensing" matrix, and the vectors are chosen randomly from this subspace.

The submitted paper did not really explain applications in which this model might arise, and several of the reviewers wondered if such applications exists. We describe three below; with two examples taken from imaging and one from covariance estimation.

1. Imaging through scattering media is a current topic of interest in the optics community, e.g. "Non-invasive imaging through opaque scattering layers" (Bertolotti et al, 2012), "Imaging With Nature: Compressive Imaging Using a Multiply Scattering Medium" (Liutkus et al, 2014). In this imaging modality, a coherent beam of light that is scattered by passing through certain semi-transparent medium is used as the source for illumination. The optical transfer function (which can be measured) of the scattering medium can be modeled by a random matrix. To create diverse structured illumination patterns, the medium is excited using an array of LEDs. Different LED excitations result in illumination patterns that are in the column space of the transfer matrix.

2. Phase retrieval with a single-pixel camera: Consider an architecture similar to the Rice University's single-pixel camera. Let x^* be a sparse vector modelling the target image. Using a conventional 4f-correlator, common in Fourier optics, we can modulate x in frequency domain by a random (say Rademacher) sequence and obtain F'DFx^*, where F is the unitary DFT matrix and D is a diagonal matrix with Rademacher diagonal entries. The resulting filtered image can be randomly modulated in the spatial domain by optical phase modulators (e.g., SLMs or DMDs) and then integrated by a lens that focuses on the single-pixel sensor. With p_i denoting the i-th spatial modulation pattern, y_i = |p_i' F'DFx^*|^2 would be the i-th (scalar) intensity measurement. If all of the spatial modulation patterns p_i are supported on the same index set S, then the observations are identical to the measurements obtained with w_i = p_i|_S and \Psi = (F')_S DF in our model. The subscript S denotes the restriction of the rows to S.

3. Estimating covariance matrices from sketches: This application is described in "Exact and Stable Covariance Estimation From Quadratic Sampling via Convex Programming" (Chen et al, 2015); the goal is to estimate the covariance matrix from compressed data vectors. Covariance matrices that are simultaneously low-rank and sparse, exactly or approximately, can be sketched using our proposed measurement model.

If accepted, we will discuss one or more of the above applications in the final version of the paper.

Simulations:

There was concern from the reviewers that the simulations are not convincing since there were only two choices of the regularization parameter. We repeated the mixed-norm minimization simulations for k=4,10, and 16 at 10 different values of \lambda between 0.1 and 9. The results, available at https://goo.gl/s8ZR9c , show that the achieved empirical error is always higher than the empirical error of the nuclear norm minimization or the \ell_1-norm minimization. This is not unexpected, given related theoretical results, i.e. Oymak et al [13]. We can include such a simulation if the paper is accepted.

2-stage method:

As Reviewer #1 pointed out, the convex programs in our 2-stage method can indeed be combined into a single convex program, by adding the objectives and enforcing the constraints jointly. However, we find it more intuitive to express the method in the 2-stage form. As the reviewer hinted, with generic measurements it may be possible to form a virtual \Psi and apply our 2-stage method. We have not analyzed this interesting scheme rigorously, but we conjecture that only an approximate solution can be guaranteed even in the noiseless scenario.
Our main contribution is a *provably efficient and robust* method for the CPR problem. Having a two-stage method is not the key point of our work. Of course, as Reviewer #6 mentioned, two-stage algorithms have already been applied to different problems.

Other comments:

Reviewer #1: Scaling the measurement vectors/matrices is equivalent to scaling the noise inversely. Thus, effectively, the (scaled) \epsilon and the SNR are reciprocally proportional.
Reviewer #2: We will revise the abstract to address the reviewers concerns, if the paper accepted.
Reviewer #3: Convex programs in each stage of our method have efficient solvers whose computational complexity are well-studied in the literature.